# Recent Advances in the Management of Pediatric Acute Myeloid Leukemia—Report of the Hungarian Pediatric Oncology-Hematology Group

**DOI:** 10.3390/cancers13205078

**Published:** 2021-10-11

**Authors:** Zsuzsanna Gaál, Zsuzsanna Jakab, Bettina Kárai, Anikó Ujfalusi, Miklós Petrás, Krisztián Kállay, Ágnes Kelemen, Réka Simon, Gergely Kriván, Gábor T. Kovács, Csongor Kiss, István Szegedi

**Affiliations:** 1Department of Pediatric Hematology-Oncology, Institute of Pediatrics, University of Debrecen, 4032 Debrecen, Hungary; gaal.zsuzsanna@med.unideb.hu (Z.G.); kisscs@med.unideb.hu (C.K.); 2National Childhood Cancer Registry, 1094 Budapest, Hungary; jakab.zsuzsanna@med.semmelweis-univ.hu; 3Department of Laboratory Medicine, University of Debrecen, 4032 Debrecen, Hungary; karai.bettina@med.unideb.hu (B.K.); ujfalusi.aniko@med.unideb.hu (A.U.); 4Division of Pediatric Hematology/Oncology, Velkey László Child’s Health Center, 3526 Miskolc, Hungary; mikey@med.unideb.hu (M.P.); kelemen.igyek@bazmkorhaz.hu (Á.K.); simonreka.igyek@bazmkorhaz.hu (R.S.); 5Pediatric Hematology and Stem Cell Transplantation Department, Central Hospital of Southern Pest, National Institute of Hematology and Infectious Diseases, 1097 Budapest, Hungary; kallay@dpckorhaz.hu (K.K.); krivan.gergely@dpckorhaz.hu (G.K.); 6Second Department of Pediatrics, Semmelweis University, 1094 Budapest, Hungary; kovacs.gabor1@med.semmelweis-univ.hu

**Keywords:** acute myeloid leukemia, acute promyelocytic leukemia, Hungarian Pediatric Oncology-Hematology Group, survival outcomes, multidimensional flow cytometry

## Abstract

**Simple Summary:**

The outcome of pediatric AML improved considerably worldwide during the past few decades. Hereby, we summarize the therapeutic results of pediatric AML patients registered between 2012 and 2019 in Hungary. As compared to our previous results, improvement was registered in event-free (EFS) and overall (OS) survival, which can be attributed to the application of contemporary diagnostic and therapeutic guidelines, advanced supportation, and higher efficacy of hematopoietic stem cell transplantation. Between 2016 and 2019, a statistically significant increment of 2-year EFS was confirmed over the period between 2012 and 2015. The most prominent progress was observed in acute promyelocytic leukemia (APL). Multidimensional flow cytometry made possible the prompt introduction of ATRA in two cases with M3v, who also represent the first pediatric APL patients in Hungary to be treated with arsenic-trioxide. Besides joining multinational pediatric AML treatment groups, our future aims include the introduction of centralized treatment centers and diagnostic facilities.

**Abstract:**

Outcome measures of pediatric acute myeloid leukemia (AML) improved considerably between 1990 and 2011 in Hungary. Since 2012, efforts of the Hungarian Pediatric Oncology-Hematology Group (HPOG) included the reduction in the number of treatment centers, contemporary diagnostic procedures, vigorous supportation, enhanced access to hematopoietic stem cell transplantation (HSCT), and to targeted therapies. The major aim of our study was to evaluate AML treatment results of HPOG between 2012 and 2019 with 92 new patients registered (52 males, 40 females, mean age 7.28 years). Two periods were distinguished: 2012–2015 and 2016–2019 (55 and 37 patients, respectively). During these periods, 2 y OS increased from 63.6% to 71.4% (*p* = 0.057), and the 2 y EFS increased significantly from 56.4% to 68.9% (*p* = 0.02). HSCT was performed in 37 patients (5 patients received a second HSCT). We demonstrate advances in the diagnosis and treatment of acute promyelocytic leukemia (APL) in two cases. Early diagnosis and follow-up were achieved by multidimensional flow cytometry and advanced molecular methods. Both patients were successfully treated with all-trans retinoic acid and arsenic-trioxide, in addition to chemotherapy. In order to meet international standards of pediatric AML management, HPOG will further centralize treatment centers and diagnostic facilities and join efforts with international study groups.

## 1. Introduction

Acute myeloid leukemia (AML) comprises approximately 25% of acute leukemia in childhood [1]. Within the 10 million population of Hungary with 10–12 new cases of pediatric AML are registered in a year. A better understanding of the complexity of pathogenesis and recent progress in defining the molecular genetic landscape of AML resulted in the development of a risk-tailored treatment approach accompanied by improved therapeutic results.

AML was a virtually incurable disease before the 1970s. Since the mid-1970s, treatment outcome results improved with a remarkable speed, yet cure rates did not reach the nearly 90% figure of pediatric acute lymphoblastic leukemia (ALL) [2,3,4]. The retrospective analysis of consecutive AML-BFM trials revealed a 49% probability of 5-year overall survival (OS) and 41% probability of 5-year event-free survival (EFS) between 1987 and 1992 [5]. Two decades later, the AIEOP group registered 68% 8-year OS and 55% 8-year EFS between 2002 and 2011 [6]. In the AML-BFM 2004 trial, 74% 5-year OS and 55% 5-year EFS were observed between 2004 and 2010 [6]. In Hungary, a country with more restricted resources than most Western European and North American countries, therapeutic results of pediatric AML also improved during the past few decades [7]. However, the increment of survival data remained to be inferior compared to the results of international trials described above. The 4-year OS increased from 34.5% to 44.8% and 47.9%, 4-year EFS increased from 24.1% to 41.4% and 45.4% between the time periods of 1990–1994, 1995–2000, and 2001–2011, respectively [8].

Improvement in international survival data was primarily associated with treatment intensification, higher efficacy of second-line treatment including allogeneic hematopoietic stem cell transplantation (HSCT), and better supportive care [5,9]. In Hungary, major problems preventing a more pronounced development included infections, in particular systemic fungal infections, bleeding, the lack of a reference diagnostic laboratory, and the decentralization of treatment (9 centers in a country of 10 million inhabitants) [8]. Since 2011, systematic efforts have been made to improve survival data of pediatric AML in Hungary. The number of treatment centers was reduced to six, including two HSCT centers. The Hungarian Pediatric Oncology-Hematology Group (HPOG) adopted the WHO diagnostic guidelines and followed the AML-BFM 2004 treatment protocol with special emphasis on supportive care and patient monitoring. Improving the efficacy of second-line treatment options, including allogeneic HSCT, was also a key objective. Here we summarize the treatment results of pediatric AML registered by HPOG between 2012 and 2019. We performed a subgroup analysis of patients with acute promyelocytic leukemia (APL) and described the last two patients of that cohort more in detail as they demonstrate the best diagnostic and therapeutic advances applied by HPOG investigators.

## 2. Materials and Methods

### 2.1. Patients

Patients aged from 0.1 to 19 years with newly diagnosed AML in Hungary between 2012 and 2019 were eligible for inclusion (*n* = 92). The diagnosis was based on current WHO guidelines, i.e., investigating MGG-stained bone marrow smears and confirmed by genetic and flow cytometry investigations as described later [10,11]. Based on the Registry of the HPOG, data of 92 children were analyzed (Figure 1).

Patients were evaluated and stratified uniformly in six Hungarian pediatric tertiary hematology-oncology centers following strictly the diagnostic guidance of the AML-BFM 2004 protocol [12]. Cheson criteria were used to define complete remission (CR), as suggested by the AML-BFM 2004 protocol [13]. In contrast to centers participating in the AML-BFM 2004 clinical trial, patients of the HPOG were not randomized and were treated according to the standard treatment arms of the protocol, i.e., “AIE” block for induction 1, “AI” block for consolidation, and 18 Gy cranial irradiation as part of CNS prophylaxis were applied. Supportive treatment guidance of the AML-BFM 2004 protocol was strictly followed, including platelet and fibrinogen supplementation, antimicrobial prophylaxis, and aggressive empiric antibiotic and antifungal therapy as well.

Informed consent was obtained for the treatment in all cases, in compliance with the corresponding ethical review board permission (IV/5639-4/2020/EKU). Baseline characteristics obtained during initial diagnostic evaluation, treatment details, and outcomes were structurally documented in the National Childhood Cancer Registry. Definitions of treatment responses were adopted from AML-BFM protocols. Early death (ED) was defined as death before the 43rd day of treatment. Death in aplasia was registered when the patient died between the 43rd and 150th day of treatment with clinical and laboratory signs of bone marrow (BM) aplasia.

### 2.2. Genetic Investigations

Bone marrow samples were cultured for 24 h in Chang culture medium according to standard protocol. Metaphase cells were stained with Giemsa (G-banding). Karyotype was described according to ISCN (2016) [14]. Fluorescent in situ hybridization (FISH) was performed according to standard procedure using XL PML-RARA Translocation/Dual Fusion Probe (Metasystems, Altlussheim, Germany). Metaphase cells and 200 interphase nuclei were analyzed using Zeiss Axioplan-2 fluorescence microscope and ISIS software (Version: V5.8.12 WK, Manufacturer: MetaSystems, Altlussheim, Germany).

Total RNA isolation was performed on EDTA-anticoagulated bone marrow samples using TRIzolate (UD-GenoMed, Debrecen, Hungary) according to standard procedure. Total RNA was reverse transcribed to cDNA using High Capacity cDNA Reverse Transcription Kit with Rnase Inhibitor (Life Technologies, Carlsbad, CA, USA) following the manufacturer protocol. In the case of *NPM1-RARA,* the RT-qPCR reaction was performed using fusion gene-specific TaqMan Assay (Thermo Fisher Scientific, Waltham, MA, USA). The *PML-RARA* fusion gene quantitation was performed using the primer sets and the PCR conditions with three primer pairs according to the Europe Against Cancer Group (EAC) protocol [15]. The level of *PML-RARA* and *NPM1-RARA* transcripts were normalized to the reference gene *ABL1* as the normalized copy number (based on the ∆C_p_ method).

*FLT3*-ITD mutation analyses were performed from genomic DNA isolated from bone marrow samples using QIAamp DNA Blood Mini Kit (Qiagen, Hilden, Germany) according to the protocol published by Buban et al. [16]. Fluorescently labeled polymerase chain reaction (PCR) products were analyzed by capillary electrophoresis using Applied Biosystems ABI 3130 Genetic Analyzer (Thermo Fisher Scientific, Waltham, MA, USA) with Genescan Analysis. The *FLT3*-ITD was given as the peak divided by the wild-type peak (ITD/WT). *FLT3*-ITD allelic ratio (AR) was calculated as the ratio of the area under the curve of mutant to wild-type alleles (*FLT3*-ITD/*FLT3*wt).

### 2.3. Flow Cytometry

During flow cytometry, eight color-labeling procedure, with a 4-tube AML panel was used to examine the bone marrow samples for diagnostic purposes. CD14, CD11b, HLA-DR, CD45, CD64, CD13, CD15, CD34, CD71, CD117, CD300e, CD4, and CD10 markers were purchased from Becton Dickinson Biosciences (San Jose, CA, USA); CD33, CD16, CD2, CD117, and CD13 markers were purchased from Beckman Coulter, (Brea, CA, USA); CD45 marker was purchased from Invitrogen (Thermo Scientific Inc., Walthman, MA, USA); HLA-DR marker was purchased from Biolegend (San Diego, CA, USA), and cyMPO was purchased from Dako (Santa Clara, CA, USA). Generation and labeling of mouse monoclonal antibodies against FXIII-A subunits were carried out as previously described using a FITC labeling kit (Sigma, St. Louis, MO, USA). A total of 100 000 events were acquired with the help of the FACS Canto II flow cytometer (Becton Dickinson Biosciences, San Jose, CA, USA). Samples were analyzed not only with the conventional method based on bivariate dot-plots but also with a novel flow cytometric analysis protocol applied multidimensional dot-plots. In the case of multidimensional dot-plot, the expression of antibodies placed in a tube and the light scatter characters (forward scatter and side scatter) were examined at the same time. This means that the expression of a certain marker on pathological cells is unknown because of the multidimensional visualization. Due to the integrated display, there is additional information about pathological cells because immunophenotype determines the position of pathological cells on multidimensional (“radar”) dot-plots. According to a previous study, multidimensional flow cytometric analysis proved to be a sensitive and specific method to identify APL cases. Pathological promyelocytes had unique positions in the multidimensional dot-plots regardless of the mechanism that led to APL. This novel protocol uses only four multidimensional dot-plots. One dot-plot was developed for each of the four tubes. During the analysis, the position of pathological cells was compared to the expected position of pathological promyelocytes in hypergranular and hypogranular APL, as defined in the previous study [17].

### 2.4. Hematopoietic Stem Cell Transplantation

Thirty-seven patients diagnosed between 2012 and 2019 received allogeneic HSCT and five patients received a second HSCT. Altogether 42 transplantations were performed. In the case of one patient, HSCT was performed in 2020 and another patient received a second HSCT in 2021. For the primary HSCTs, different conditioning regimens were applied including busulphan-cyclophosphamide-melphalan (Bu/Cy/Mel, *n* = 2), busulphan-cyclophosphamide-etoposide (Bu/Cy/Ve, *n* = 1), busulphan-fludarabine (Bu/Flu, *n* = 15), busulphan-fludarabine-melphalan (Bu/Flu/Mel, *n* = 4), busulphan-clofarabine-melphalan (Bu/Clo/Mel, *n* = 1), busulphan-clofarabine-fludarabine (Bu/Clo/Flu, *n* = 1), busulphan-clofarabine-cyclophosphamide (Bu/Clo/Cy, *n* = 1), fludarabine-thiotepa-melphalan (Flu/Thio/Mel, *n* = 2), treosulfan-fludarabine (Treo/Flu, *n* = 2), treosulfan-fludarabine-thiotepa (Treo/Flu/Thio, *n* = 7) and treosulfan-fludarabine-melphalan (Treo/Flu/Mel, *n* = 1). Except three cases of Bu/Flu and one case of Treo/Flu/Mel conditioning, anti-thymocyte globulin (ATG) was also administered. Conditioning regimens for retransplantations applied were: Bu/Flu/ATG (*n* = 1), Treo/Flu/Thio/ATG (*n* = 2), Flu/Thio/Mel (*n* = 1), and TBI/Ve (*n* = 1; performed in 2021). Types of grafts were bone marrow (BM), cord blood unit (CBU) and peripheral blood stem cell (PBSC) in 17, 5, and 20 cases, respectively. Types of donors included matched unrelated donors (MUD, *n* = 36), haploidentical donors (*n* = 3), matched related (sibling) donors (*n* = 2) and an identical twin donor (*n* = 1). Of the second HSCTs, all the grafts were PBSCs and all donors were MUDs.

### 2.5. Statistical Analysis

Statistical analysis of data was performed with the SigmaStat 3.0 Software (Version: 3.0, Manufacturer: Systat Software, San Jose, CA, USA). Survival probabilities were estimated using the Kaplan–Meier method and Statistica 7.0 Software (Version 7, Manufacturer: StatSoft, Tulsa, OK, USA). Based on the non-normal distribution of data, a nonparametric Mann–Whitney test was performed for the comparison of groups of patients. *p* values of <0.05 were considered statistically significant.

## 3. Results

Between 2012 and 2019, a total of 92 cases was registered with de novo AML in Hungary. Among the evaluated 92 cases, there were 52 males and 40 females with ages ranging from 0.1 to 19 years (mean 7.28 yrs). Detailed clinicopathological characterization of patients is given in Table 1. Four of the patients were diagnosed with a coexisting Down syndrome, among which three SR patients are still alive in CR1.

All patients were treated uniformly according to the AML-BFM 2004 protocol. In the total 2012–2019 cohort of patients, CR was 83/92 (90.2%), the 2 y and 5 y overall survival (OS) were 64.4% and 56%, respectively; and the 2 y and 5 y event-free survival (EFS) were 57.6% and 49.7%, respectively (Figure 2a). In the SR group of patients, 2 y OS was 93.0%, and 5 y OS was 79.9%, while 2 y EFS was 85.8%, and 5 y EFS was 72.5%. In the HR group of patients, 2 y OS was 51.8%, and 5 y OS was 45.4%, while 2 y EFS was 44.2%, and 5 y EFS was 37% (Figure 2b). Cumulative incidence of relapse (CIR) and disease-free survival (DFS) data are also included (Table 2).

Although the small number of individual recurrent genetic aberrations did not allow a statistical evaluation of treatment outcome measures according to genetic subgroups, there was a clear difference in treatment outcome measures of the SR vs. the HR group of patients, which stratification considered strongly, among other clinicopathological factors, the presence of favorable vs. other recurrent genetic aberrations (Figure 2b). Moreover, there was a remarkable difference in the outcome of patients representing the most frequent favorable and unfavorable genetic subgroups. t(8; 21) was registered in nine cases, among which eight patients reached CR and are still in CR1 (median follow-up time 5.5 yrs). Among patients with myelodysplasia-related changes, −7/−7q was the most frequent genetic aberration (*n* = 8), and five of these eight patients died.

During the total study period (2012–2019), the number of adolescent patients (age > 14 years) was 15 (12/15 HR and 3/15 SR patients). The CR rate was 73.3%. The rate of relapse among patients achieving CR was 54.5% (6/11). Eight of the 15 adolescent patients have died.

During the 2012–2015 period (Period I) and the 2016–2019 period (Period II), 55 and 37 patients were analyzed, respectively. Sex and age distribution of the two groups of patients did not differ significantly: 32 males and 23 females in Period I vs. 20 males and 17 females in Period II (*p* = 0.694) with a mean age of 7.1 yrs (Period I) vs. 7.6 years (Period II) (*p* = 0.712). The distribution of standard risk (SR) and high risk (HR) cases also did not differ significantly with 15 and 14 SR and 40 and 23 HR patients in Period I and Period II, respectively (*p* = 0.37).

In Period I, the rate of remission was 49/55 (89.1%), the 2 y OS was 63.6%, and the 2 y EFS was 56.4% (Figure 2c). The relapse rate was 12/55 (21.8%). Of the patients who were diagnosed in Period I, 26 patients died. Four cases of ED were registered: three patients died before the 15th day of induction treatment, and one patient died between the 15th and 43rd day of induction. Four patients died in aplasia. Death in CR after 150 days was registered in seven cases. The number of patients who died after relapse was 11.

The rate of remission in Period II exhibited an improving trend with 34/37 (91.9%). Survival data in Period II have improved considerably compared to Period I with a nearly significant elevation of 2 y OS of 71.4% (*p* = 0.057) and with a statistically significant increment of 2 y EFS of 68.9% (*p* = 0.02) (Figure 2c). The relapse rate in Period II was 11/37 (29.7%), which did not differ significantly from Period I (*p* = 0.501). Of the patients who were diagnosed in Period II, 11 patients died. Two cases of ED were registered between the 15th and 43rd day of induction. Three patients died in aplasia. Death in CR after 150 days was registered in one case. Five patients died after relapse.

The most important causes of death were infection and progression in both analyzed time periods (Table 3).

The number of patients who died due to infection decreased from 12/55 to 3/37 within Period I and Period II, respectively, though this difference was not statistically significant (*p* = 0.279). Here we note that the number of lethal fungal infections decreased from four registered cases in Period I to zero during Period II. Progression was registered as the cause of death in eight and seven cases in Period I and Period II, respectively. In both periods, more than half of these patients died after relapse (five patients in Period I and four patients in Period II). Comparing Period I and Period II, decreasing number of deaths was observed during CR after 150 days (7/55 and 1/37, respectively); however, this difference was not statistically significant (*p* = 0.426). One patient with hyperleukocytosis syndrome developed fatal bleeding during leukapheresis (Period I). ED of one patient with acute megakaryoblastic leukemia was attributed to acute cardiotoxicity (Period I).

### 3.1. Patients Undergoing HSCT

Among patients diagnosed between 2012 and 2019, 37 underwent HSCT (1/37 HSCT was performed in 2020), and 5 of them received a second HSCT (1/5 performed in 2021). Of the HSCTs, 23 were carried out in Period I, and 19 were performed in Period II. The number of transplantations was 24, 13, 1, 2, and 2 in CR1, CR2, CR3, PR, and PD, respectively. Among the above-described types of grafts and donors, PBSCs (*n* = 20) and matched unrelated donors (*n* = 36) were the most frequent entities. Death following HSCT was registered in 51.4% of patients (*n* = 19). Of those five patients receiving a second HSCT, three patients are alive in CR. The number of the HSCT-related death in Period I was 14, while it was 5 in Period II. Graft versus host disease (GvHD) developed in 29.7% of patients (*n* = 11), involving the skin in all cases and the gut in six and the liver in four cases. Among the patients with GvHD, PBSCT was performed in eight cases, BMT in three patients, and six of the patients with GvHD died. Causes of deaths after GvHD involved multiorgan failure (*n* = 4), Aspergillus infection (*n* = 1), relapse (*n* = 1) and progressive disease (PD, *n* = 1).

### 3.2. Patients with Acute Promyelocytic Leukemia

Between 2012 and 2019, 10 pediatric cases of APL were registered in Hungary with five-five patients in Period I and II. Diagnosis was based on morphological investigation of the diagnostic bone marrow smears and was confirmed by flow cytometry and genetic investigations. Eight patients were presented with characteristic APL morphology (AML M3), and two patients were diagnosed with AML M3v. RARA rearrangement was identified in 8/10 cases: seven patients exhibited t(15;17), and one patient had t(5;17). One further patient had a normal karyotype, and the genetic diagnosis was not successful in one case. All 10 patients with APL received a combined cytostatic treatment according to the AML-BFM 2004 protocol. All patients with confirmed RARA rearrangement (8/10) received 25 mg/m^2^/day of all-trans retinoic acid (ATRA) started on the day of diagnosis (day 1 of treatment). ATRA was given in 14 day-long blocks once each month during the treatment. Two patients with AML M3v also received arsenic-trioxide (ATO) in the course of their treatment, as described below in more detail. During Period I, three APL patients achieved CR, and two died. Causes of death were pulmonary mucormycosis and fatal gastrointestinal bleeding. In Period II, all of the five APL patients achieved CR and are alive.

Morphological evaluation of two patients with AML M3v was not unequivocal. In these cases, advanced flow cytometry methods helped in establishing the proper diagnosis. Here we describe two cases of children with AML M3v in more detail, where multidimensional flow cytometry provided early evidence of APL allowing the initiation of adequate treatment. Moreover, these two cases represent the first application of ATO in Hungarian patients.

### 3.3. Clinical Presentation

#### 3.3.1. Case 1

An 8-month-old Caucasian female infant was admitted to the tertiary pediatric treatment center because of a rapidly growing mass over the mandibular arch. CBC suggested the presence of acute leukemia (Hb 70 g/L, Ht 0.20 L/L, RBC 2.32 T/L, Plt 98 G/L, WBC 56.2 G/L) with atypical blast cells in the peripheral blood. Biopsy from the mandibular lesion and bone marrow aspiration was performed. In the bone marrow smear, there were 32% atypical non-erythroid blast cells. The blasts were medium-sized cells with folded nuclei and basophilic cytoplasm that did not contain granules. Auer rods were not present (Figure 3a). Morphology raised the possibility of either AML M3v or AML M5 requiring further laboratory evaluation.

#### 3.3.2. Case 2

A 6-year-old Caucasian boy was referred to the tertiary pediatric treatment center with isolated thrombocytopenia (Hb 113 g/L, Ht 0.31 L/L, RBC 4.1 T/L, Plt 32 G/L, WBC 8.37 G/L upon referral) and superficial skin bleedings suggesting immune thrombocytopenia. He was admitted for watch-and-wait. However, rapidly increasing WBC during the next two days (15 G/L and 23.5 G/L, respectively) suggested acute leukemia and bone marrow aspiration was performed. With the presence of 39% non-erythroid blast cells, the morphologic picture was similar to that observed in case one (Figure 3b), requiring diagnostic confirmation by additional laboratory methods.

### 3.4. Flow Cytometry

The pathological promyelocytes of patient 1 exhibited atypical immunophenotype by conventional flow cytometry analysis. These cells expressed myeloid markers such as CD13, CD33, cyMPO, and they did not, or only partially expressed monocytic markers such as CD64, CD4, CD14, and cyFXIII-A. The pathological cells did not express blast markers, such as CD34, HLA-DR, and TdT, and expressed certain markers that are usually present in myeloid cells in a later stage of development, such as CD16 and CD10. In summary, conventional flow cytometry analysis, based on bivariate dot-plots, suggested that these cells belonged to the myeloid lineage, but the immunophenotype of the pathological cells was not typical for APL or another subtype of AML. In Case 2, results of conventional flow cytometry analysis suggested hypogranular type of APL. Pathological cells expressed myeloid markers, the expression of cyMPO was bright. They were CD117-positive and CD34-, HLA-DR-, and CD15-negative. However, results of the multidimensional flow cytometric analysis suggested unequivocally that both patients had APL. More than 95% of the pathological cells were detected in the APL gate pre-defined in our previous study in all multidimensional dot-plots in both cases (Figure 4) [17]. This means that the positions of pathological cells were typical for APL. In addition, histologic evaluation of the mandibular mass revealed the presence of granulocytic sarcoma in Case 1.

### 3.5. Genetics

Genetic examination confirmed APL in both cases by karyotyping and FISH. Case 1 exhibited the very rare recurrent variant translocation 46,XX,t(5;17)(q35;q21)(13)/46,XX(7).nuc ish(PMLx2,RARAx3)(185/200) (Figure 5a,b). RT-qPCR analysis confirmed the presence of the *NPM1-RARA* fusion gene with high expression level (404.18%). Case 2 showed the “classical” t(15;17)(q22;q21) translocation. His karyotype was 46,XY,t(15;17)(q22;q21)(19)/46,XY(1).nuc ish(PML,RARA)x3(PML con RARAx2)(178/200). The *PML-RARA* fusion gene (short type) expression level was 32.65%. In addition, the leukemic clone of patient 2 exhibited *FLT3*-ITD mutation, as revealed by fluorescent PCR. The inserted fragment length was 27 bp, the AR 0.64.

### 3.6. Treatment and Outcome

CR was successfully achieved in both cases with combinatorial administration of AML-BFM 2004 protocol and ATRA. ATRA was started on day 1 in both cases based on the results of multidimensional flow cytometry, and it was applied in a dose of 25 mg/m^2^/day, orally given in two single doses, with 14 days treatment on and 7 days off, and so on, until remission was achieved. None of the two patients developed any kind of hemostatic complications either during induction or in later treatment phases. Patient 2 developed mild symptoms of elevated central nervous system pressure during ATRA blocks in the maintenance phase of the treatment. Evaluation of cerebrospinal fluid did not reveal central nervous system involvement. Symptoms completely disappeared upon application of dexamethasone and mannitol. Suspension or dose tapering of ATRA was not necessary. ATO was started in the maintenance phase of anti-leukemic treatment in both cases, parallel with the routine continuous 40 mg/m^2^ 6-thioguanin plus monthly 40 mg/m^2^ cytarabine for 4 days, at a dose of 0.15 mg/kg/day, 2 weeks on and 1 month off, for a total of 10 cycles, applying ATRA at 25 mg/m^2^ during these cycles as well, without significant side effects. Minimal residual disease was checked on a regular basis in both cases by flow cytometry and RT-qPCR (Table 4). At present, both patients are alive in CR1 with a follow-up time of 36 and 20 mo, respectively.

## 4. Discussion

Improving survival results of pediatric AML is a complex task that requires elaborated efforts even from large national and multinational consortia operating in industrialized, high-income countries. Recent clinical trials executed by such large pediatric hematology working groups reported on remission and survival rates at around 90% and 70%, respectively [6,18]. In addition to contemporary diagnostic workup, anti-leukemic treatment, and supportive therapy, healthcare settings also have a considerable impact on survival outcomes. A recent cross-national study described that the overall risk of mortality from childhood AML was 23% lower in the U.K. as compared to that in the USA; therefore, a critical evaluation of health care programs is also needed to uncover the underlying mechanisms and to close these gaps in mortality [19].

Hungary, a country with restricted resources as compared to the most developed European and North American economies, has demonstrated that pediatric AML management can be successfully improved, approaching results of leading pediatric oncology groups [7,18]. Comparing the presently investigated period of 2012–2019 with results of a report of the previous period of 2001–2011 showed that CR increased from 47.3% to 57.0%, and survival rates increased from 47.9% to 56.0% [8]. More importantly, treatment results improved from the first period of the present investigation, i.e., 2012–2015, and the second period of 2016–2019. CR, 2 yr OS, and 2 yr EFS improved from 50%, 63.6%, and 56.4% in Period I to 67,8%, 71.4%, and 68.9% in Period II, respectively. These results are approaching treatment outcome measures of international and multicentric pediatric AML trial groups, although they do not reach results of pediatric AML working groups in similar-sized Western European countries, such as Austria and the Netherlands [19,20,21]. From the region of the Central-Eastern European countries, only data from the Polish group were reported. The most recent interim Polish results with 5 yr OS and 5 yr EFS of 63% and 51.8%, respectively, are similar to the results of the Hungarian group [22].

Improvements achieved by the Hungarian group can be attributed to a number of factors. First, the number of treatment centers was reduced from 10 to 6. This way, a more experienced team of pediatric hematologists and nurses was able to manage children with this relatively rare disease. Second, diagnostic measures according to current WHO guidelines were followed [10,11]. Flow cytometry was performed in every case, and the majority of patients had a genetic diagnosis (Table 1). Third, treatment guidance, including vigorous supportive therapy of the AML-BFM 2004 protocol, was strictly followed in each treatment center. Although infectious death was still one of the leading causes of mortality after disease progression, yet the ratio of patients succumbing to infections, in particular to systemic fungal infections, was considerably smaller in the present period of 2012–2019 than in the previous period reported [8]. In Period II, more patients received novel antifungal agents such as echinocandine derivatives and liposomal amphotericin B, as compared to Period I, when antifungal treatment was started with fluconazol, and in case of microbiologically proven fluconazol resistance or in case of poor clinical response, conventional amphotericin B was applied. In addition, the biggest study site was moved to a new building by Period II. Death to major bleeding occurred only in 2/92 cases, whereas it was a major factor of mortality in the period of 2001–2011 [8].

HSCT was applied more frequently (37/92 patients) than in the previous period of 2001–2011 (30/112 patients). The majority of grafts originated from matched unrelated donors in the present period. The ratio of the HSCT-related death showed a remarkable decrease when comparing the data of Period I (number of deaths: 14) to Period II (number of deaths: 5), that is, an improvement from 14/23 = 60.8% to 5/19 = 26.3% was seen, when analyzing the survival data after HSCTs in these time periods.

The most prominent progress was observed among patients with APL. Diagnosis was ascertained by demonstrating the specific translocations. The introduction of multicolor flow cytometry resulted in definitive diagnosis within a few hours allowing the start of ATRA therapy on day one, together with combined chemotherapy. Although a uniform APL immunophenotype does not exist, conventional flow cytometry based on the evaluation of bivariate dot-plots and morphological evaluation of bone marrow smears is usually sufficient for diagnosing typical cases of APL. The definition of AML M3v is, however, challenging. Morphological evaluation of bone marrow smears of patients with AML M3v may result in misleading diagnoses, such as AML M4, M5, or JMML. The immunophenotype of APL cells is often unusual [23]. Our group has recently developed and published a new technique where identification and characterization of APL cells were based on evaluating multidimensional dot-plots. Implementation of this method allows the easy and accurate definition of APL cells, including variant and cryptic cases, within a short turn-around time [17]. Application of advanced diagnostic measures made possible the prompt introduction of ATRA in each case in the investigated period. In accordance with the results of Testi et al., who administered ATRA in the same dose (25 mg/m^2^/day) as our group, no clinically significant side effects were registered, ATRA was generally well tolerated by patients [24]. In the APL group of patients, there was 1/10 fatal gastrointestinal bleed, and a further patient succumbed to pulmonary mucormycosis. Eight/10 children with APL are in CR1 and alive.

In the two cases of children with M3v, prompt diagnosis and start of ATRA treatment were made possible by multidimensional dot-plot evaluations of the immunophenotype. APL rarely occurs in the infant age group [24,25]. To the best of our knowledge, Case 1 represents the first infant with APL M3v due to the variant translocation of t(5;17). The new transcriptional variant t(5;17)(q35;q21), resulting in the formation of the NPM1-RARA fusion gene, was first described in 1996 [26]. A 6-mo-old male infant with cutaneous mastocytosis and aleukemic leukemia cutis was reported earlier harboring an identical translocation. In that case, however, bone marrow myeloid cells exhibited a normal morphology despite the abnormal karyotype, and the case resolved spontaneously after 14 months [27]. Variant APL blasts with t(5;17) have been reported to respond to ATRA by differentiation [28]. The case described here, together with the other case of APL M3v with t(15;17), responded favorably to combined treatment with ATRA and cytostatic drugs. In addition, these two pediatric APL cases were the first ones in Hungary to be treated with ATO. Experiences with the application of ATO in pediatric APL are limited, and its role and place in pediatric APL treatment protocols have not yet been exactly defined. However, observations suggest that it can be used more favorably in combination with ATRA and chemotherapy than in monotherapy, as pointed out by the presented two cases, as well [24,29,30].

This study has some limitations. First, patients in the observational period were investigated retrospectively. Second, because of the low incidence of pediatric AML, the restricted number of patients treated in a country of about 10 million inhabitants did not allow a statistical analysis detailed to the minute. The small number of individual recurrent genetic aberrations, the number of patients with Down syndrome, and the number of adolescent patients (age > 14 years) prevented the statistical evaluation of treatment outcome measures of these subgroups. *FLT3*-ITD has been analyzed only recently in two study sites; therefore, the impact of *FLT3*-ITD was not evaluated. Finally, technical advances made it possible to monitor and to follow-up MRD only in the case of the last two patients with APL included in this investigation.

## 5. Conclusions

In conclusion, reasonable reorganization of health care provision and application of contemporary diagnostic and therapeutic guidelines may result in considerable progress in the management of pediatric AML even in countries with relatively restricted resources, such as Hungary. Despite the limited size of the studied population, we demonstrated a progressive improvement in the treatment results of pediatric AML between 2012 and 2019 as compared with previous periods. In pediatric APL, the implementation of innovative diagnostic and therapeutic approaches contributed to the favorable outcome in the majority of patients, including those with M3v. Based on its clearly reported strong prognostic impact [31,32,33] and our experiences obtained with AML M3v, HPOG decided to analyze MRD prospectively by flow cytometry in all patients with AML in the course of the coming protocol. Further progress can be expected by further reduction in the number of treatment centers attending children with AML and by establishing a central laboratory facility. Better use of collaborative efforts will be aimed at by joining to currently forming multinational pediatric AML treatment groups.

## Figures and Tables

**Figure 1 cancers-13-05078-f001:**
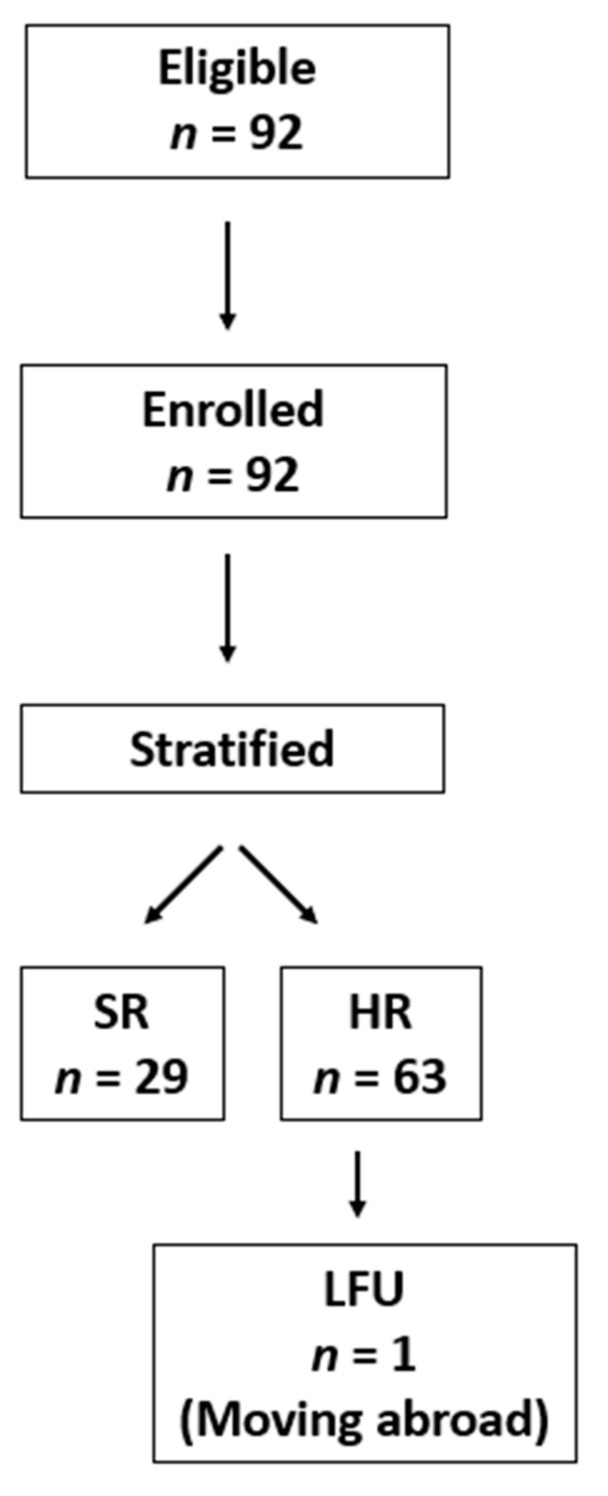
Flow of participants from enrollment to follow-up and analysis. Patients of the HPOG were not randomized and were treated according to the standard treatment arms of the protocol, i.e., “AIE” block for induction 1, “AI” block for consolidation, and 18 Gy cranial irradiation as part of CNS prophylaxis were applied. Abbreviations: HR: high risk, LFU: lost to follow-up, SR: standard risk.

**Figure 2 cancers-13-05078-f002:**
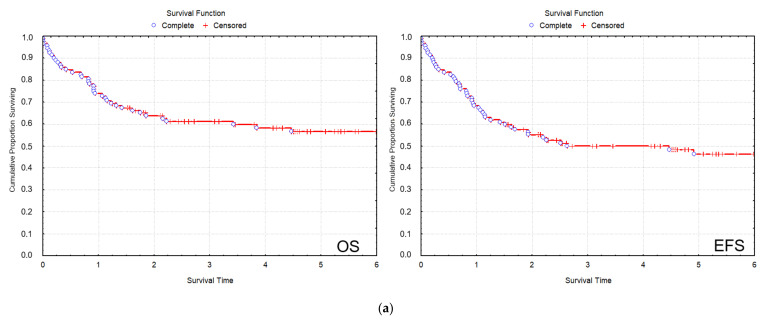
(**a**) Kaplan–Meier analysis of the OS and EFS values in the total cohort of patients between 2012 and 2019. (**b**) OS and EFS curves of SR and HR group of patients. (**c**) OS and EFS of patients registered in Period I (2012–2015) and Period II (2016–2019). (5 y OS of Period I and Period II was 52% and 67%, respectively, *p* < 0.05, Cox F test; 5 y EFS of Period I and Period II was 44% and 55%, respectively, *p* = 0.105, Cox F test). Abbreviations: EFS: event-free survival, HR: high risk, OS: overall survival, SR: standard risk.

**Figure 3 cancers-13-05078-f003:**
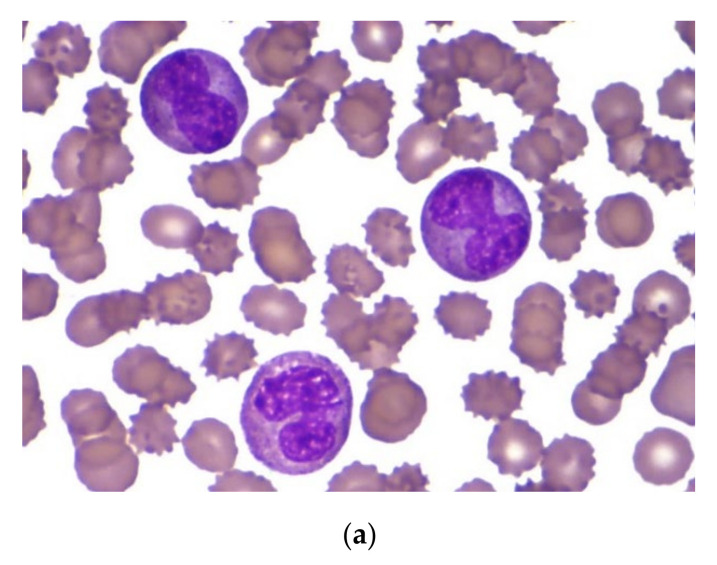
(**a**) Morphology of pathologic cells in the bone marrow smear in case 1. (**b**) Morphology of pathologic cells in the bone marrow smear in case 2.

**Figure 4 cancers-13-05078-f004:**
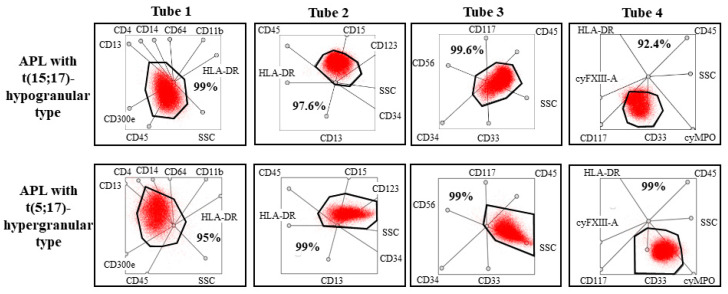
Multidimensional dot-plots of patients with APL. More than 95% of blasts were detected in the pre-defined gate in the cases of the two patients with classic and variant translocation of the RARA gene.

**Figure 5 cancers-13-05078-f005:**
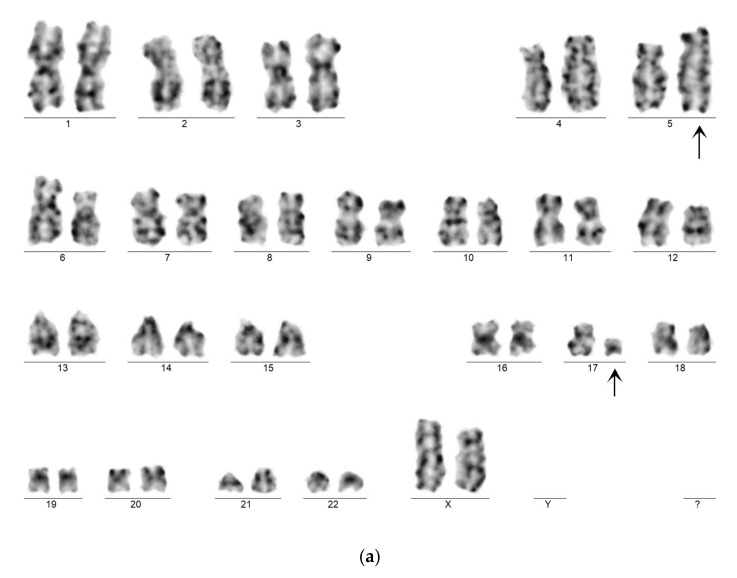
(**a**) Representative karyogram of t(5;17)(q35;q21) translocation. Arrows show the derivative chromosomes taking part in the translocation. (**b**) An interphase nucleus with two copies of PML signal (red) and three copies of RARA signal (green) indicating the translocation of RARA gene. Abbreviations: PML: promyelocytic leukemia protein, RARA: retinoic acid receptor alpha.

**Table 1 cancers-13-05078-t001:** Clinicopathological characterization of pediatric AML patients registered in the Hungarian Childhood Tumor Register between 2012 and 2019.

Number of Registered Patients	Recurrent Genetic Abnormalities	92
Sex		
Male		52
Female		40
Age at the time of the diagnosis (years)		
0–9		57
10–14		20
15–19		15
Leukocytes		
<10		35
10–100		43
>100		14
Risk group		
SR		29
HR		63
CNS stage		
CNS1		82
CNS2		4
CNS3		6
FAB classification		
M0		16
M1		9
M2		18
M3		10
M4		16
M5		18
M6		0
M7		5
WHO classification (2008)		
Acute myeloid leukemia (AML) with recurrent genetic abnormalities		31
	AML with t(8;21)(q22q22.1)	9
	AML with inv(16)(p13.1q22)	5
	APL with PML-RARA	6
	AML with t(9;11)(p21.3;q23.3)	5
	AML with t(6;9)(p23;q34.1)	1
	AML with inv(3)(q21.3q26.2) or t(3;3)(q21.3;q26.2)	2
	AML with t(1;22)(p13.3;q13.1)	1
	AML with mutated NPM1	2
	AML with mutated CEBPA	0
AML with myelodysplasia-related changes		15
Therapy-related myeloid neoplasms		0
AML, not otherwise specified (NOS)		30
No data		16

Abbreviations: AML: acute myeloid leukemia, APL: acute promyelocytic leukemia, CNS: central nervous system, FAB: French-American-British classification, HR: high risk, NOS: not otherwise specified, SR: standard risk, WHO: World Health Organization.

**Table 2 cancers-13-05078-t002:** Summary of outcome analysis.

Groups of Patients		2 y OS	5 y OS	2 y EFS	5 y EFS	CR	Relapse	CIR 2 yrs	DFS 2 yrs	DFS 5 yrs	Death
Total cohort (*n* = 92)		64.4%	58%	55%	46%	83/92 (90.2%)	23/92 (25%)	20.70%	70.1% (SE 5.4%)	60.9% (SE 6.1%)	37/92 (40.2%)
Risk groups	SR (*n* = 29)	93%	80%	86%	73%	25/29 (86.2%)	5/29 (17.2%)	15% ^a^	85.80% ^b^	69.90%	5/29 (17.2%)
	HR (*n* = 63)	52%	45%	44%	37%	58/63 (92.1%)	18/63 (28.6%)	24% ^a^	60.60% ^b^	55.30%	32/63 (50.8%)
Periods	Period I	63%	52%	54%	44%	49/55 (89.1%)	12/55 (21.8%)	19%	70.90%	60.20%	26/55 (47.3%)
	Period II	67%	67%	63%	55%	34/37 (91.9%)	11/37 (29.7%)	22%	69.20%	64.70%	11/37 (29.7%)

Abbreviations: CIR: cumulative incidence of relapse, CR: complete remission, DFS: disease-free survival, EFS: event-free survival, OS: overall survival. ^a^ and ^b^ indicate statistically significant differences between the corresponding groups (*p* < 0.05).

**Table 3 cancers-13-05078-t003:** Deaths according to treatment and classification of causes of deaths in Period I and Period II.

Loss of Patients According to Treatment Phases and to Causes of Death	Period I	Period II
Death according to treatment period		
early death <15 days	3	0
early death 15-42 days	1	2
death in aplasia 43–150 days	4	3
death in CR after 150 days	7	1
death after relapse	11	5
total number of deaths	26	11
CR after induction	49/55 (89.1%)	34/37 (91.9%)
Primary induction failure	6/55 (10.9%)	3/37 (8.1%)
Infection		
bacterial	1	0
fungal	4	0
viral	1	0
not identified	6	3
total	12	3
Bleeding	1	1
Progression		
progression of de novo AML	2	1
progression after relapse	6	6
total	8	7
Toxicity	2	0
Late organ injury	1	0
Other cause	2	0

**Table 4 cancers-13-05078-t004:** MRD monitoring in two AML M3v cases. The level of PML-RARA and NPM1-RARA transcripts were normalized to the reference gene ABL1 as the normalized copy number (based on the ∆C_p_ method).

Treatment Day	Day 1	Day 1	Day 1	Day 25	Day 25	Day 25	Day 42	Day 88 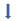	Day 88	Day 112	Day 140
Method of MRD detection	BM FC	FISH	qPCR	BM FC	FISH	qPCR	BM FC	FISH	qPCR	qPCR	qPCR
Patient 1	82%	93%	404%	49%	46%		28%	2%		0.01%	neg
Patient 2	77%	89%	32.6%	0.06%		0.04%	0.005%		neg	neg	neg

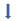
: start of arsenic-trioxide treatment. Abbreviations: BM FC: bone marrow flow cytometry; FISH: fluorescent in situ hybri-dization; MRD: minimal residual disease; qPCR: quantitative polymerase chain reaction.

## Data Availability

The data presented in this study are available on request from the corresponding author. The data are not publicly available due to privacy.

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
