# Peer review of "Recent Advances in the Management of Pediatric Acute Myeloid Leukemia—Report of the Hungarian Pediatric Oncology-Hematology Group"

_cancers, 2021, doi:10.3390/cancers13205078_

Round 1
Reviewer 1 Report
The manuscript is an original article on the treatment results of pediatric AML registered in the Hungarian Pediatric Oncology Group between 2012 and 2019. 92 pediatric AML patients were treated in 6 centers according to the AML-BFM 2004 protocol and the results are here illustrated. The paper also includes the report of 10 APL and a focus on two peculiar cases. The presentation of the results is clear even if can be improved.
Major issues:
- The paper is composed by a first section on the treatment results of the HPOG and a second on the description of two cases of APL. Although the two cases are really interesting the two section of the manuscript seems too far to coexist in a single paper considering that the first part is a retrospective cohort study and the second a case series. I would recommend shorten the focus on the APL and further expanding the cohort description.
- Where patients with Down Syndrome included in the analysis? If so, I would recommend adding a specification also reporting, if possible, the outcomes.
- In the outcome analysis should be reported also the complete remission rate, the cumulative incidence of relapse, the relapse rate, the CI of death in CR and the disease free survival. These should be done for the whole population and according by risk groups.
- The criteria used for the risk group allocation should be more clearly defined in the text.
- A clear part in which are reported the complete outcome data (overall survival, event free survival, disease free survival, cumulative incidence of relapse and cumulative incidence of death in complete remission) according to the major recurrent genetic aberrations identified, is not present in the manuscript. This should be added and maybe illustrated in a figure or a table.
- How is the CR defined? Do the authors analyze the MRD status during the treatment? If so, it would be of extreme interest to see whether the relationship between MRD and outcome is confirmed in this cohort (PMID: 28240765).
- A flow chart of the patient distribution during treatment would be useful, as in many clinical trial or master protocol.
- How many primary induction failure were reported? and how is the quote of PIF according by risk groups.
- How was the ED according by the two different subgroups
Minor issues:
- In Table 1 I would recommend adding the presence of CNS leukemia (Y/N) and FLT3 status. Moreover, I would recommend adding data divided by period with related statistics to assess whether baseline characteristics of the two periods were different. This specification would help to better interpretate the data.
- I would suggest the authors consider the addition in Table 2.a of the number and percentage of patients in CR after induction, primary induction failure and deaths divided by study periods.
- Considering the interesting results in acute lymphoblastic leukemia (PMID: 22010101) I would suggest analyzing the data comparing treatment results for children (<14 y) and AYA (14-19 y).
- Figure 1c.: it would be useful to add the p value inside the Kaplan-Meyer curve. Figure quality should be improved. Censored cases should be reported.
- Can the authors better comment upon the strong reduction of fungal infections between the two periods? Did antifungal prophylaxis or treatment change between the two periods?
Author Response
The manuscript is an original article on the treatment results of pediatric AML registered in the Hungarian Pediatric Oncology Group between 2012 and 2019. 92 pediatric AML patients were treated in 6 centers according to the AML-BFM 2004 protocol and the results are here illustrated. The paper also includes the report of 10 APL and a focus on two peculiar cases. The presentation of the results is clear even if can be improved.
Major issues:
- The paper is composed by a first section on the treatment results of the HPOG and a second on the description of two cases of APL. Although the two cases are really interesting the two section of the manuscript seems too far to coexist in a single paper considering that the first part is a retrospective cohort study and the second a case series. I would recommend shorten the focus on the APL and further expanding the cohort description.
Yes, we agree with the reviewer that publishing evaluation and interpretation of treatment results in parallel with the description of case reports can be considered unusual. However, we did not put these two entities of editing approaches in parallel to each other. The second section of the manuscript is the summary of patients with APL in a similar way as we summarized all the AML cases. This main section includes APL cases analyzed more in detail in the second section. Out of this subgroup analysis, we described the two most recent cases which demonstrate best diagnostic and therapeutic advances, i.e. the application of multi-dimension flow-cytometry analysis, MRD monitoring by RT-qPCR, and the application of arsenic trioxide in addition to combined cytostatic and ATRA treatment. Moreover, we think it is worth mentioning that the infant with t(5;17) presenting as an APL M3v, represents the first case published. To stress our aim and to explain the somewhat unusual compilation of the manuscript, we added a last sentence to „Introduction”: „We performed a subgroup analysis of patients with acute promyelocytic leukemia (APL) and described the last two patients of that cohort more in detail as they demonstrate best diagnostic and therapeutic advances applied by HPOG investigators.” (Lines 95-98 in the revised manuscript.)
- Where patients with Down Syndrome included in the analysis? If so, I would recommend adding a specification also reporting, if possible, the outcomes.
To address reviewer’s question we added a last sentence to the first paragraph of „Results” (lines 219-220 in the revised manuscript):
“Four of the patients were diagnosed with a coexisting Down syndrome, among which three SR patients are still alive in CR1”
- In the outcome analysis should be reported also the complete remission rate, the cumulative incidence of relapse, the relapse rate, the CI of death in CR and the disease free survival. These should be done for the whole population and according by risk groups.
In order to summarize results of the outcome analysis, an additional table was inserted in the section of Results (Table 2). Please note, that due to this additional table, the numbering of the further tables has been changed.
|
|
|
2y OS |
5y OS |
2y EFS |
5y EFS |
CR |
Relapse |
CIR 2 yrs |
DFS 2 yrs |
DFS 5 yrs |
Death |
|
Total cohort (n=92) |
|
64.4% |
58% |
55% |
46% |
83/92 (90.2%) |
23/92 (25%) |
20,70% |
70,1%(SE 5,4%) |
60,9%(SE 6,1%) |
37/92 (40.2%) |
|
Risk groups |
|
|
|
|
|
|
|
|
|
|
|
|
|
SR (n=29) |
93% |
80% |
86% |
73% |
25/29 (86.2%) |
5/29 (17.2%) |
15%* |
85,80%* |
69,90% |
5/29 (17.2%) |
|
|
HR (n=63) |
52% |
45% |
44% |
37% |
58/63 (92.1%) |
18/63 (28.6%) |
24%* |
60,60%* |
55,30% |
32/63 (50.8%) |
|
Periods |
|
|
|
|
|
|
|
|
|
|
|
|
|
Period I |
63% |
52% |
54% |
44% |
49/55 (89.1%) |
12/55 (21.8%) |
19% |
70,90% |
60,20% |
26/55 (47.3%) |
|
|
Period II |
67% |
67% |
63% |
55% |
34/37 (91.9%) |
11/37 (29.7%) |
22% |
69,20% |
64,70% |
11/37 (29.7%) |
Table 2. Summary of outcome analysis. Abbreviations: CIR: cumulative incidence of relapse, CR: complete remission, DFS: disease-free survival, EFS: event-free survival, OS: overall survival
*indicates statistically significant difference between the corresponding groups.
- The criteria used for the risk group allocation should be more clearly defined in the text.
Risk group allocation was performed according to the stratification described in the AML-BFM 2004 protocol [Creutzig, U.; Zimmermann, M.; Bourquin, J.P.; Dworzak, M.N.; Fleischhack, G.; Graf, N.; Klingebiel, T.; Kremenes, B.; Lehrnbecher, T.; von Neuhoff, C. et al. Randomized trial comparing liposomal daunorubicin with idarubicin as induction for pediatric acute myeloid leukemia: results from Study AML-BFM 2004. Blood. 2013. 122(1):37-43].
To address this remark of the Reviewer, we changed the 5th sentence in the first paragraph of “Materials and Methods” and added two new sentences and reference items after the modified sentence (Lines 113-120 in the revised manuscript):
“Patients were evaluated and stratified uniformly in six Hungarian pediatric tertiary hematology-oncology centers following strictly the diagnostic guidance of the AML-BFM 2004 protocol [12]. Cheson criteria were used to define complete remission (CR), as suggested by the AML-BFM 2004 protocol [13]. In contrast to centers participating in the AML-BFM 2004 clinical trial, patients of the HPOG were not randomized and were treated according to the standard treatment arms of the protocol, i.e. “AIE” block for induction 1, “AI” block for consolidation, and 18 Gy cranial irradiation as part of CNS prophylaxis were applied.”
Please note that the insertion of the new references items resulted in a change in the numbering of citations.
- A clear part in which are reported the complete outcome data (overall survival, event free survival, disease free survival, cumulative incidence of relapse and cumulative incidence of death in complete remission) according to the major recurrent genetic aberrations identified, is not present in the manuscript. This should be added and maybe illustrated in a figure or a table.
Due to the small number of cases within distinct subgroups of patients according to recurrent genetic aberrations, statistical analysis could not be performed to compare outcome data of these groups. To address this remark, we inserted a last paragraph in the „Discussion” listing the shortcomings of this research: „This study has some limitations. First, patients in the observational period were investigated retrospectively. Second, because of the low incidence of pediatric AML, the restricted number of patients treated in a country of about 10 million inhabitants did not allow a statistical analysis detailed to the minute. In particular, the small number of individual recurrent genetic aberrations prevented the statistical evaluation of treatment outcome measures according to genetic subgroups. Finally, technical advances made possible to monitor and to follow-up MRD only in case of the last two patients with APL included in this investigation.”
Although the small number of individual recurrent genetic aberrations did not allow a statistical evaluation of treatment outcome measures according to genetic subgroups, there was a clear difference in treatment outcome measures of the SR vs. the HR group of patients which stratification considered strongly, among other clinicopathological factors, the presence of favorable vs. other recurrent genetic aberrations (Figure 2b). Moreover, there was a remarkable difference in the outcome of patients representing the most frequent favorable and unfavorable genetic subgroups. t(8;21) was registered in nine cases, among which eight patients reached CR and are still in CR1 (median follow-up time 5.5 yrs). Among patients with myelodysplasia-related changes, -7/-7q was the most frequent genetic aberration (n=8), and five of these eight patients died.
We inserted these sentences to the end of the 2nd paragraph of „Results” (lines 240-249 in the revised manuscript).
- How is the CR defined? Do the authors analyze the MRD status during the treatment? If so, it would be of extreme interest to see whether the relationship between MRD and outcome is confirmed in this cohort (PMID: 28240765).
CR was defined according to Cheson criteria as suggested by the AML-BFM 2004 (see response to remark No. 4).
MRD was analyzed only in the last two case of APL as pointed out in the newly added last paragraph of „Discussion” (see response to remark No. 5). Nevertheless, we completely agree with you that investigating the relationship between MRD and outcome would be of extreme interest. Therefore, we plan to investigate this issue prospectively in the coming period. To highlight the importance of your suggestion, we inserted a sentence into „Conclusions” (lines 559-561 in the revised manuscript): „Based on the experiences obtained with AML M3v, HPOG decided to analyze MRD prospectively in tall patients with AML in course of the coming protocol.”
- A flow chart of the patient distribution during treatment would be useful, as in many clinical trial or master protocol.
To address this request, a flow of participants was inserted into the Methods section (Figure 1), therefore numbering of the further figures have changed.
- How many primary induction failure were reported? and how is the quote of PIF according by risk groups.
CR was not achieved in nine cases. In the total 2012-2019 cohort of patients, CR was 83/92 (90,2%) (line 228 of the revised manuscript).
Four patients belonged to the SR group (4/29) and five patients (5/63) belonged to the HR group. As it is indicated in Table 2, CR was 25/29 (86.2%) in the SR group, and CR was 58/63 (92.1%) in the HR group of patients.
- How was the ED according by the two different subgroups
ED occured in six cases. All of these patients belonged to the HR group.
In Period I, four cases of ED were registered: three patients died before the 15th day of induction treatment and one patient died between the 15th and 43rd day of induction (lines 263-265) of the revised manuscript). In Period II, two cases of ED were registered between the 15th and 43rd day of induction (lines 290-291) of the revised manuscript).
Minor issues:
- In Table 1 I would recommend adding the presence of CNS leukemia (Y/N) and FLT3 status. Moreover, I would recommend adding data divided by period with related statistics to assess whether baseline characteristics of the two periods were different. This specification would help to better interpretate the data.
We have included CNS status in Table 1.
FLT3-ITD investigation was introduced only recently and only in two of the six study sites. Therefore, only 24 patients were investigated regarding to FLT3 status (FLT3-ITD mutation was confirmed in six cases. and 18 cases were FLT3-ITD negative), and this is why we did not perform a separated evaluation according to FLT3 status.
- I would suggest the authors consider the addition in Table 2.a of the number and percentage of patients in CR after induction, primary induction failure and deaths divided by study periods.
These data are included in Table 3 in the revised version on page 11 (numbering of tables have changed due to one additional table inserted in the revised version).
- Considering the interesting results in acute lymphoblastic leukemia (PMID: 22010101) I would suggest analyzing the data comparing treatment results for children (<14 y) and AYA (14-19 y).
During the total study period (2012-19), the number of adolescent patients (age > 14 years) was 15 (12/15 HR and 3/15 SR patients). The CR rate was 73.3%. The rate of relapse among patients achieving CR was 54.5% (6/11). Eight of the 15 adolescent patients have died.
- Figure 1c.: it would be useful to add the p value inside the Kaplan-Meyer curve. Figure quality should be improved. Censored cases should be reported.
p values are included in the legend of Figure 2c in the revised version (please note, that numbering of figures has been changed due to one additional figure).
- Can the authors better comment upon the strong reduction of fungal infections between the two periods? Did antifungal prophylaxis or treatment change between the two periods?
In Period II, more patients received novel antifungal agents such as echinocandine derivatives and liposomal amphotericin B, as compared to Period I, when antifungal treatment was started with fluconazol, and in case of microbiologically approven fluconazol resistance or in case of poor clinical response, conventional amphotericin B was applied. In addition, the biggest study site moved to a new building by Period II.
These sentences were inserted in the Discussion in lines 490-495.

Reviewer 2 Report
In this work, authors evaluated AML treatment results of HPOG between 2012 and 2019 demonstrating an improvement in that period thanks to implementation of innovative diagnostic and therapeutic approaches.
I think some minor adjustments need to be made.
In particular:
- Always put the abbreviation in full the first time it appears in the text.
- Not split the figures, tables and captions in two pages but put entirely on the same page accompanied by their caption.
- Legend to table 1: please listed ALL the abbreviations, i.e., SR, HR, FAB, WHO, etc…
- I think it would be better to prepare a single figure 1 including figures 1A, 1b and 1c, with a single caption.
- The same is for figure 2 (2a and 2b): merge them or numbered them separately (i.e., figure 2 and figure 3). Again for figure 4a and 4b.
- Table 2a and 2b should be better separated with more obvious edge or merge them into a single table. In table 2b, the words bacterial, fungal, viral, not identified must be aligned. I think the ∑ symbol is not necessary.
Author Response
In this work, authors evaluated AML treatment results of HPOG between 2012 and 2019 demonstrating an improvement in that period thanks to implementation of innovative diagnostic and therapeutic approaches.
I think some minor adjustments need to be made.
Thank you for your recommendations and for reviewing our manuscript. As detailed below, we have included the suggested modifications in the revised version.
In particular:
- Always put the abbreviation in full the first time it appears in the text.
The location of the full versions of the abbreviations APL and CR have been corrected. Acute promyelocytic leukemia (APL) can be found in line 96 and complete remission (CR) is in line 116 in the revised version.
- Not split the figures, tables and captions in two pages but put entirely on the same page accompanied by their caption.
All tables and figures are put entirely on the same page in the revised version of the manuscript. In order to address this remark, font size in Table 1 was reduced.
- Legend to table 1: please listed ALL the abbreviations, i.e., SR, HR, FAB, WHO, etc…
Abbreviations are listed in the legend of Table 1 in the revised version as the following:
Abbreviations: AML: acute myeloid leukemia, APL: acute promyelocytic leukemia, CNS: central nervous system, FAB: French-American-British classification, HR: high risk, NOS: not otherwise specified, SR: standard risk, WHO: World Health Organization (lines 225-228 in the revised manuscript)
- I think it would be better to prepare a single figure 1 including figures 1A, 1b and 1c, with a single caption.
Due to one additional figure and one additional table, the numbering of all figures and tables have been changed. Figure 2 was composed as a single figure in the revised version on page 10.
- The same is for figure 2 (2a and 2b): merge them or numbered them separately (i.e., figure 2 and figure 3). Again for figure 4a and 4b.
Due to one additional figure and one additional table, the numbering of all figures and tables have been changed. Figures 3 and 5 have also been prepared as single figures in the revised version of the manuscript (pages 14 and 16, respectively).
- Table 2a and 2b should be better separated with more obvious edge or merge them into a single table. In table 2b, the words bacterial, fungal, viral, not identified must be aligned. I think the ∑ symbol is not necessary.
Due to one additional figure and one additional table, the numbering of all figures and tables have been changed. The original Table 2a and 2b were merged into a single table (Table 3 on page 11 in the revised manuscript). The listed words are aligned and the ∑ symbol is replaced with the word „total”.

Round 2
Reviewer 1 Report
The authors fully addressed many of the issues raised and the manuscript improved. The outcome analysis has been better clarified.
In the conclusion when it has been added the phrase "„Based on the experiences obtained with AML M3v, HPOG decided to analyze MRD prospectively in all patients with AML in course of the coming protocol.” I think that it should be better to stress the value of MRD evaluation by flow citometry in the forthcoming new protocols of treatment. This because this evaluation was clearly reported having a strong prognostic impact in some important papers that the authors should cite (see PMID: 31681710, PMID: 20451454 and PMID: 16877738)
Author Response
Here we would like to express our gratefulness for the useful comments and questions.
To emphasize the importance of flow cytometry in MRD analysis, we have modified the 4th sentence in Conclusions as the following:
„Based on its clearly reported strong prognostic impact [31, 32, 33] and our experiences obtained with AML M3v, HPOG decided to analyze MRD prospectively by flow cytometry in all patients with AML in course of the coming protocol” - lines 559-562 of the revised manuscript.
According to the PMID numbers listed, three new reference items were included in the manuscript– references 31, 32 and 33.
Spell check was also performed, based on which the following corrections were made:
-line 475: similar-sized instead of similar sized
-line 493: proven instead of approven
-line 504: multicolor instead of multi-color
-line 561: all patients instead of tall patients